# High-Precision Temperature Inversion Algorithm for Correlative Microwave Radiometer

**DOI:** 10.3390/s21165336

**Published:** 2021-08-07

**Authors:** Jie Liu, Kai Zhang, Jingyan Ma, Qiang Wu, Zhenlin Sun, Hao Wang, Youquan Zhang

**Affiliations:** Faculty of Information Technology, Beijing University of Technology, Beijing 100124, China; KaiZhang@emails.bjut.edu.cn (K.Z.); majy0405@emails.bjut.edu.cn (J.M.); qiangwu@bjut.edu.cn (Q.W.); dmahz13@163.com (Z.S.); haowang@bjut.edu.cn (H.W.); S201961721@emails.bjut.edu.cn (Y.Z.)

**Keywords:** correlative radiometer, correction algorithm, integral calibration, PSO-LM-BP inversion algorithm

## Abstract

In order to achieve high precision from non-contact temperature measurement, the hardware structure of a broadband correlative microwave radiometer, calibration algorithm, and temperature inversion algorithm are innovatively designed in this paper. The correlative radiometer is much more sensitive than a full power radiometer, but its accuracy is challenging to improve due to relatively large phase error. In this study, an error correction algorithm is designed, which reduces the phase error from 69.08° to 4.02°. Based on integral calibration on the microwave temperature measuring system with a known radiation source, the linear relationship between the output voltage and the brightness temperature of the object is obtained. Since the metal aluminum plate, antenna, and transmission line will have a non-linear influence on the receiver system, their temperature characteristics and the brightness temperature of the object are used as the inputs of the neural network to obtain a higher accuracy of inversion temperature. The temperature prediction mean square error of a back propagation (BP) neural network is 0.629 °C, and its maximum error is 3.351 °C. This paper innovatively proposed the high-precision PSO-LM-BP temperature inversion algorithm. According to the global search ability of the particle swarm optimization (PSO) algorithm, the initial weight of the network can be determined effectively, and the Levenberg–Marquardt (LM) algorithm makes use of the second derivative information, which has higher convergence accuracy and iteration efficiency. The mean square error of the PSO-LM-BP temperature inversion algorithm is 0.002 °C, and its maximum error is 0.209 °C.

## 1. Introduction

The research and development of non-contact temperature measurement equipment is of great significance for medical clinical application, especially in emergency department and intensive care units (ICUs) [1,2]. Non-contact temperature measurement equipment can detect hot spots in subcutaneous tissue of ICU patients with a depth of 3 cm to 4 cm and a diameter of 2 cm. Non-contact temperature measurement can be carried out on the human body, which lays a good foundation for preventing the spread of the coronavirus disease 2019 (COVID-19). It has become the development trend of non-contact temperature measuring equipment to develop a thermometer that measures temperature with high accuracy even under clothing [3].

In 2011, Gilreath et al. used a five-stage low-noise amplifier to optimize the W band direct detection receiver in the best condition and minimize the deviation [4]. In 2013, Zhang et al. improved the structure of the superheterodyne receiver by dividing the converted signal into two orthogonal I/Q signals, which were sampled and uploaded to the computer for analysis and processing [5]. Both of the above research studies used a diode for detection, which caused a large error when the input power is small. In this paper, the concept of correlative detection is applied to the design of the receiver, which can effectively limit the influence of uncorrelated signal and noise component [6,7].

To improve the temperature measurement accuracy, the high-precision temperature inversion algorithm is carried out based on measurements from the correlative microwave radiometer [8,9]. In recent years, a series of achievements have been made in the field of temperature inversion algorithms. In 2016, Lou et al. proposed an inversion method of an internal temperature field of slow wave structure based on a BP network model for the measurement of internal temperature of a traveling-wave tube [10]. However, since it is easy for the BP network to fall into local optimal solution, further optimization is needed to achieve higher accuracy. In 2019, Ling et al. used the particle swarm optimization (PSO) algorithm to optimize the BP network for the inversion of the magnetic nanoparticles’ temperature measurement model, and Ling et al. combined the Gaussian Newton iteration method (GN) with PSO to obtain the PSO-GN algorithm. This algorithm improves the local search ability of the temperature inversion algorithm, but the effect is poor under the condition of low SNR, and the error is greater than that of the traditional PSO algorithm [11].

Using a correlative microwave radiometer after phase correction and integral calibration, a linear relationship between receiver output voltage and brightness temperature can be obtained. The high-precision PSO-LM-BP inversion algorithm is innovatively proposed in this paper to predict the temperature, which uses the global search ability of the PSO algorithm to determine the initial weight and then uses the second-derivative information of an LM algorithm to obtain high accuracy. The microwave temperature measuring system is introduced from the following four aspects: (1) structure of microwave radiometer temperature measuring system, focusing on the radio frequency front end, orthogonal demodulator, difference amplifier, and analog-to-digital converter; (2) the phase correction algorithm corrects the I/Q channel signal phase error and amplitude maladjustment caused by the original signal passing through the power divider; (3) signal after autocorrelation arithmetic, the square of the input voltage contained in the output voltage, so the direct linear relation between the output voltage and input power—on this basis, the integral calibration is to determine the output voltage and brightness temperature linear mathematical expression, according to a two-point calibration method, using radiation characteristics of two known sources, this article for absorbing material and an aluminum drum. By determining the linear mathematical expression, the brightness temperature of the measured object corresponding to the output voltage can be obtained; (4) interference factors are added to the input end of the neural network to eliminate the interference on an antenna caused by the environment, receiver, and aluminum barrel. The improved PSO-LM-BP neural network inversion algorithm is implemented on the corrected and calibrated data to get the predicted temperature value with high accuracy. The process of correlative microwave radiometer calibration and temperature inversion is shown in Figure 1.

## 2. Materials and Methods

### 2.1. Structure of Microwave Radiometer Temperature Measuring System

The design block diagram of the microwave receiver is shown in Figure 2. The signal is processed by a multi-stage low-noise amplifier and a multi-stage bandpass filter; in this design, three low-noise amplifiers (ZX60-83LN-S+, Mini-Circuits, Brooklyn, NY, USA) with noise figures of 1.59 dB and gain of 20 dB are placed at the front end of the radio frequency front end, which can effectively reduce the noise figure of the receiver system. Then, we cascade three low-noise amplifiers with two 15 dB low-noise amplifiers (ZX60-V63+, Mini-Circuits) to make the radio frequency front end gain up to 90 dB. At the same time, a set of bandpass filters is added between the low-noise amplifiers to control the bandwidth of the receiver. Each bandpass filter is composed of a high-pass filter (VHF-3800+, Mini-Circuits) and a low-pass filter (VLF-5500+, Mini-Circuits); then, the receiver autocorrelation element reaches the working point. In order to realize autocorrelation, the I+,I−/Q+,Q− channel signals are realized by the power divider and multiplier. After the differential amplifier and AD converter, the output voltage signal represents the power value of the measured signal. The amplitude of the output signal of the complex correlator is proportional to the square of the voltage amplitude of the received signal, and the phase of the output signal is the phase difference of the two received signals, so the linear relationship between the input power and the output voltage can be established [12]. In addition, the radiation power received by the antenna has a linear relationship with the brightness temperature of the measured target [13]. Therefore, based on experimental data and the supervised neural network algorithm, the microwave receiver can determine the physical temperature of the object. Applying the concept of a complex correlator to the structure design of a receiver cymoscope can not only realize the function of microwave radiation power detection but also effectively reduce the influence of system noise.

The actual temperature measuring system is built as shown in Figure 3a. The receiver designed in this paper mainly consists of a radio frequency front end, orthogonal demodulator, difference amplifier, and analog-to-digital converter. The correlative microwave receiver designed in this paper will be used to measure the temperature of human skin and subcutaneous tissue [14,15]. Therefore, the microwave receiver should have a wide working frequency bandwidth from 4 to 6 GHz. Considering the non-contact method of human body temperature detection, this paper designed a pyramid horn antenna for the microwave radiometer front end in order to reduce the impact of environment on the system noise. Its advantages are low-voltage standing wave ratio, wide working bandwidth, high directivity, and easy to manufacture. The technical specifications of the antenna are as follows: the width of the horn is 88.04 mm, the depth of the horn is 89.05 mm, and the length of the horn is 49.08 mm. The phase accuracy and amplitude balance index of the orthogonal demodulator should be high enough to ensure that the input signal can be recovered from the output result after the correlation calculation. In this paper, ADL5380-EVALZ from Analog Devices is selected, which can operate in the bandwidth range of 400 MHz to 6 GHz, and it has high phase accuracy and amplitude balance, which can effectively guarantee the accuracy of the receiver. The fully differential amplifier amplifies the signal after autocorrelation processing. In order to ensure the accuracy, ADA4940-2ACP-EBZ fully differential dual-channel amplifier from Analog Devices is selected in this paper. Its misadjustment error is 0.35 mV and the rail-to-rail output is −VS + 0.1 V to +VS − 0.1 V, which has good dynamic performance. EVAL-AD7903SDZ from Analog Devices is used as part of the analog-to-digital converter to obtain the digital quantity of the signal. It consists of two channels and uses a successive approximation conversion mode with high speed, low power loss, and no delay. By matching the EVAL-SDP-CB1Z evaluation board, the analog-to-digital converter sampling data can be uploaded to a PC for analysis and processing.

This paper uses a model with electromagnetic properties similar to human tissue for physical temperature measurement. The permittivity model parameters of various tissues show that muscle, blood, and skin have high water content [16,17], so this paper uses water as the measurement object to study the performance of a non-contact temperature measurement system based on a correlative receiver. The test diagram is shown in Figure 3b. Next, this paper will discuss the correction, calibration, and inversion algorithm of the wideband correlative microwave radiometer for non-contact temperature measurement.

### 2.2. Design of Correction Algorithm for Correlative Microwave Receiver

In practical measurement, the output voltage of the I/Q channel correlative microwave receiver is as follows:(1)$$V_{I}=\left(hB^{2}+e\right)\cos\left(\varphi \right)\;$$
(2)$$V_{Q}=\left(hB^{2}+e\right)\sin\left(\varphi \right).$$

After the power divider, φ is the system phase error caused by the circuit; e and h are maladjustment; and B is the output signal amplitude of the power divider.

The amplitude and phase of the output voltage are:(3)$$\;V=\sqrt{V_{I}{}^{2}+V_{Q}{}^{2}}=\sqrt{{\left(\left(hB^{2}+e\right)\cos\left(\varphi \right)\right)}^{2}+{\left(\left(hB^{2}+e\right)\sin\left(\varphi \right)\right)}^{2}}=hB^{2}+e$$
(4)$$\arctan\left(\frac{V_{Q}}{V_{I}}\right)=\arctan\left(\frac{\left(hB^{2}+e\right)\sin\left(\varphi \right)}{\left(hB^{2}+e\right)\cos\left(\varphi \right)}\right)=\varphi$$

The 16-bit AD converter is used in this paper, the sampling time is 2 min, and a total of 2048 groups of data are measured. The main parameters obtained are the code value after the AD sampling I/Q channel and the current object temperature value. The results of system partially sampling data collation are shown in Table 1.

The receiver is connected to the standard signal source, and the phase shifter is controlled to a 22.5° step scan phase; then, we record the phase shift of the phase shifter, collect the data of the I/Q channel sampling values at each phase point, and calculate the average voltage value and average phase value of the phase point. The phase error is the absolute value of phase mean minus theoretical phase. The measurement results of the uncorrected phase sweeping part are shown in Table 2.

The total phase error of the system can be obtained by averaging the phase error of the sampled data, which is 69.08°, and then correcting the phase error in the software through the correction algorithm. The software correction phase error is shown in Equations (5) and (6).
(5)$$V_{Ichannel}=V_{max}\left[\cos(\theta_{TARGET})\cos\left(\varphi_{PHASE_{shift}}\right)-\sin(\theta_{TARGET})\sin\left(\varphi_{PHASE_{shift}}\right)\right]$$
(6)$$V_{Qchannel}=V_{max}\left[\sin(\theta_{TARGET})\cos\left(\varphi_{PHASE_{shift}}\right)+\cos(\theta_{TARGET})\sin\left(\varphi_{PHASE_{shift}}\right)\right]$$

The scanning phase at 22.5° step size, the original sampling data of the I/Q channel, and the data after phase error correction by the software algorithm are shown in Figure 4.

Figure 4 shows that after fitting the optimal curve, there is a linear relationship between the voltage value and the sampling value after phase compensation. The intercept and slope of the fitting line are the offset e and the slope $h_{real}$, respectively. The ideal misalignment of each signal chain in the receiver subsystem should be 0 for the least significant bit. The slope of the optimal fitting line represents the slope of the subsystem. The ideal subsystem slope can be calculated as follows:(7)$$h_{\mathrm{ideal}\;}=\frac{Code_{max\;}-Code_{min\;}}{+V_{\mathrm{REF}}--V_{\mathrm{REF}}}$$
where $Code_{max\;}$is the maximum sampling value; $Code_{min\;}$is the minimum sampling value; the reference voltage is 5 volts; and $V_{\mathrm{REF}}$ is 5 V. In this paper, a 16-bit AD converter is used, and the digital value range after conversion is 0–65,535. The error offset contains the DC component, so the corrected offset is the original offset minus the DC component, which can be expressed as:(8)$$e_{\mathrm{correct}}=e-32768$$

The gain error correction coefficient is:(9)$$h_{\mathrm{correct}\;}=\frac{h_{\mathrm{ideal}\;}}{h_{real\;}}$$

The actual received sample value after correction:(10)$$Code_{correct}=Code*h_{correct}+e_{correct}$$

According to the error correction in Equation (10), the gain error and offset error can be corrected. The results are shown in the Table 3.

Table 3 shows that the mean phase error of the system is reduced to 4.02°. In the actual measurement of water temperature, the mean phase error can be reduced to 1.4° by using the correction algorithm in the temperature range of 37.5 to 38 °C. The correction algorithm can effectively reduce the influence of phase error, gain error, and offset error on the output results.

### 2.3. Calibration Based on the Temperature Measuring System

Calibration is to receive the radiation energy of the calibration source with known radiation characteristics through the antenna and record the output voltage of the receiver at this time. Calibration is divided into step calibration [18] and integral calibration, which is a two-point integral calibration method widely applied in microwave radiometer calibration, so this paper designs an integral calibration scheme with an absorbing material and aluminum drum for the non-contact temperature measurement system composed of a correlative microwave receiver and high directivity antenna, in which the receiver and antenna are calibrated at the same time.

Integral calibration is used to determine the output voltage and brightness temperature linear mathematical expression, which is established in the process of signal after autocorrelation arithmetic, including the square of the input voltage contained in the output voltage, so featuring the direct linear relation between the output voltage and input power [19,20,21]. According to a two-point calibration method, using radiation characteristics of two known sources, one article for an absorbing material and an aluminum drum, the brightness temperature of the measured object corresponding to the output voltage can be obtained.

In this paper, two calibration sources with different emissivity, the metal aluminum barrel and absorbing material, are selected to design the integral calibration scheme. The emissivity of the absorbing material is close to 1 and can be regarded as a blackbody. The surface of the metal aluminum plate is in contact with air and has an oxide film. According to relevant literature [22,23] and historical experience, its emissivity can be approximately 0.26. Real objects are shown in Figure 5, and the relevant parameters are shown in Table 4.

After the brightness temperature of the radiation source and the corresponding receiving voltage at the same brightness temperature are known, the calibration formula determined by the two-point calibration method is as follows:(11)$$V=0.0019776*T_{B}+1.1991173.$$

Due to the interference of the electromagnetic environment, the loss of equipment components, and the change of detection position, it is necessary to calibrate regularly. Generally, the integral calibration is carried out with absorbing material in the environment of liquid nitrogen and at room temperature. In order to solve the problem of difficult storage and preparation of absorbing material in a liquid nitrogen environment, this paper uses a metal aluminum drum instead of it. The calibration scheme in this paper meets the needs of the calibration process.

### 2.4. Inversion Algorithm of Object Temperature Prediction

In 2021, Sun et al. used a BP neural network inversion algorithm to predict the temperature; the average error is 0.575 °C and the maximum error is 1.358 °C [24]. This result did not meet the accuracy standard of medical temperature measurement (±0.2 °C) [25]. In this paper, an improved PSO-LM-BP neural network inversion algorithm is proposed to predict the temperature with high precision.

Three digital thermometers with metal probes are used to measure the temperature of the aluminum drum, the antenna surface, and the transmission line. All of them may affect the inversion results. The metal aluminum drum will accumulate the energy emitted by itself into the antenna temperature in the form of radiation. The rise of antenna interface temperature and transmission line temperature will affect the measurement results by affecting the output voltage of the receiver. Different factors will lead to different effects [26,27]. By using a neural network, the inversion algorithm is a non-linear inversion problem. At the input end of the neural network, interference factors are added, which is characterized by temperature, to eliminate the interference on the antenna caused by the environment, receiver, and aluminum barrel. Using the improved PSO-LM-BP neural network structure, the actual temperature of the object can be retrieved with high accuracy.

The unified parameters used in the following networks are as follows:Brightness temperature, aluminum barrel temperature, antenna temperature, transmission line temperature, and water temperature are taken as the inputs of the supervised neural network.The training set and validation set are 124 sets of data and 30 sets of data in the temperature range from 27.5 to 64.5 °C. Each algorithm is validated in five rounds and the validation set is selected at random.BP network structure: The four input values are normalized, the number of network layers is three, the number of hidden layer neurons is seven, and the output layer outputs the predicted temperature.Activation function f(x):(12)$$f\left(x\right)=\frac{1}{1+e^{-x}}$$
where x is the input to the activation function. The non-linear function is introduced as the activation function, so that the expression ability of the deep neural network is more powerful, and the output is no longer a linear combination of the input, but it can approximate almost any function.Error function E(x):(13)$$E\left(x\right)=\frac{1}{2}\overset{N}{\underset{i=1}{\sum }}{\left(Y_{i}-T_{i}\right)}^{2}=\frac{1}{2}\overset{N}{\underset{i=1}{\sum }}e_{i}{}^{2}\left(x\right)$$
where $Y_{i}$ is the neural network predicted value, $T_{i}$ is the true value, N is the number of training set groups, and i is the number of each group.

#### 2.4.1. BP Neural Network Algorithm

The BP neural network does not need to determine the mapping relationship between the input and output in advance. Through the process of forward propagation and error back propagation, the network constantly adjusts the weights and thresholds of the network to minimize the error between the actual output value and the expected output value. Good robustness, adaptability, fault tolerance, and generalization have made the BP neural network algorithm one of the important algorithms in the field of artificial intelligence [28,29,30].

In this section, the parameters of the BP neural network are as follows: the number of iterations is 2000, and the learning rate is 0.01.

For 30 validation sets, the experimental data show that the average error of the BP neural network algorithm is 0.824 °C, the maximum error is 3.351 °C, and the mean square error is 0.629 °C. The large deviation of the BP neural network algorithm in retrieving the water temperature is due to:The structure of the BP neural network only using the gradient descent method, which makes it easy to fall into a local minimum.The accuracy of temperature sensors for measuring the temperature of the aluminum plate, antenna surface, and transmission line is limited, which leads to the low accuracy of data used in the inversion algorithm.

Aiming at the problem that it is easy for the BP neural network water temperature inversion algorithm to fall into the local minimum, this paper designs and uses particle swarm optimization (PSO) and the Levenberg–Marquardt (LM) algorithm to optimize the BP neural network without changing the network structure.

#### 2.4.2. PSO-BP Neural Network Algorithm

The particle swarm optimization (PSO) algorithm is based on the foraging behavior of birds, which is an optimal decision-making process. Each particle in the population has two attributes: velocity and position. Each particle has a position and the velocity of particles creates the weight matrix in a BP neural network. There is only one initialization weight matrix in a BP algorithm, and the PSO-BP neural network algorithm can initialize more matrices, increasing the global search ability of the network. The velocity and position of the particles are brought into the BP forward neural network to calculate and compare with the historical fitness value, and then, the individual optimal value is determined. Through comparison, the individual with the best fitness value is selected as the global optimal value, which is used to guide and update all the particle velocities. The specific PSO-BP neural network algorithm process is shown in Figure 6.

In the process of back propagation, the traditional BP algorithm uses the gradient descent method to update the weight and threshold parameters. When the BP algorithm falls into a flat area, the weight change becomes smaller, resulting in slow convergence or even failure of convergence. Using the PSO algorithm to optimize the BP neural network algorithm can combine the advantages of the two algorithms [31,32]. The updated formula of weight in the PSO algorithm depends on the performance of all particles, which does not depend on the gradient information, which increases the global search ability of the network and effectively prevents the BP algorithm from falling into the local minimum value.

In this section, the parameters of the PSO-BP neural network are as follows: the number of iterations is 500; the number of particles is 20; the range of velocity is (−5, 5) and the range of position is (−10, 10); the acceleration constant is 2; and the linear weight decreasing strategy is used.

The specific iterative formula of particle position and velocity is as follows:(14)$$v_{i+1}=w_{k}*v_{i}+c_{1}*rand_{1}*\left(pbest_{i}-x_{i}\right)+c_{2}*rand_{2}*\left(gbest_{i}-x_{i}\right)$$
(15)$$x_{i+1}=x_{i}+v_{i+1}.$$

$w_{k}$ adopts linear weight decreasing strategy:(16)$$w_{k}=\frac{\left(w_{ini}-w_{end}\right)*\left(T_{max}-k\right)}{T_{max}}+w_{end}$$
where $T_{max}$ is the maximum number of iterations.$\;c_{i}$ is the acceleration constant. $rand_{i}$ is a random number between (0, 1).$\;v_{i}$ and $x_{i}$ represent the velocity and position of the *i*-th dimension of a particle, respectively. $gbest_{i}$ represents the weight of the particle with the best fitness among all the particles in the *i*-th dimension, and $pbest_{i}$ is the weight corresponding to the historical optimal solution of the particle itself in the *i*-th dimension. k is the number of iterations, and the inertia coefficient decreases linearly from 1 to 0 as the number of iterations increases.

For 30 validation sets, the experimental data show that the average error of the PSO-BP neural network algorithm is 0.521 °C, the maximum error is 2.920 °C, and the mean square error is 0.210 °C. The back propagation process optimized by the PSO algorithm can effectively improve the problem in which it is easy for the BP neural network water temperature inversion algorithm to fall into a local minimum.

#### 2.4.3. LM-BP Neural Network Algorithm

The Levenberg–Marquardt (LM) algorithm is a non-linear optimization method combining the gradient descent method and Gauss–Newton method. The algorithm adds a factor μ. When the solution is far away from the optimal solution, the LM-BP algorithm is equivalent to the gradient descent method and has the ability of global judgment. When the distance is close to the optimal solution, the LM-BP algorithm is equivalent to the Gauss–Newton method and has the ability of local judgment. By using the second derivative information of the Newton method, the problem is that the gradient descent method drops slowly or even fails to converge when it is close to the minimum point, which is effectively alleviated. By using the Jacobian matrix in the Gauss Newton method, the problem in which Newton’s method needs a lot of calculation in a Hessian matrix is solved.

The error function with matrix increment is:(17)$$E\left(x+\Delta x\right)=\frac{1}{2}e^{T}\left(x+\Delta x\right)e\left(x+\Delta x\right),\;e\left(x+\Delta x\right)=e\left(x\right)+J\left(x\right)\Delta x$$
where E(x) is the error matrix, and x is the matrix composed of the weight and threshold.

Jacobian matrix elements are shown in Equation (18). $e_{i}\left(x\right)$ represents the error of the i-th group of input vectors after passing through the weight matrix.
(18)$$J\left(x\right)=\left[\begin{array}{ccc} \begin{array}{cc} \frac{\partial e_{1}\left(x\right)}{\partial x_{1}} & \frac{\partial e_{1}\left(x\right)}{\partial x_{2}}\\ \frac{\partial e_{2}\left(x\right)}{\partial x_{1}} & \frac{\partial e_{2}\left(x\right)}{\partial x_{2}} \end{array} & \cdots & \begin{array}{c} \frac{\partial e_{1}\left(x\right)}{\partial x_{k}}\\ \frac{\partial e_{2}\left(x\right)}{\partial x_{k}} \end{array}\\ \vdots & \ddots & \vdots \\ \begin{array}{cc} \frac{\partial e_{n}\left(x\right)}{\partial x_{1}} & \frac{\partial e_{n}\left(x\right)}{\partial x_{2}} \end{array} & \cdots & \frac{\partial e_{n}\left(x\right)}{\partial x_{k}} \end{array}\right]$$

It is necessary to satisfy the error function first derivative equal to 0, so that E(x+Δx) can reach minimum; the updated matrix increment is:(19)$$\Delta x=\underset{\Delta x}{\min}\left\{E\left(x+\Delta x\right)\right\}=-{\left[J^{T}\left(x\right)J\left(x\right)\right]}^{-1}J^{T}\left(x\right)e\left(x\right).$$

The above is the derivation of the Gauss–Newton method. The LM-BP algorithm is adopted after increasing the coefficient μ:(20)$$\Delta x=-{\left[J^{T}\left(x\right)J\left(x\right)+\mu I\right]}^{-1}J^{T}\left(x\right)e\left(x\right),\;\;\;x_{i+1}=x_{i}+\Delta x.$$

Using the LM algorithm to optimize the BP algorithm can effectively improve the convergence speed of the BP algorithm, and its algorithm diagram is shown in Figure 7.

In this section, the parameters of the LM-BP neural network are as follows: the number of iterations is 300; the minimum gradient value $g_{min}$ is $10^{-10}$; step parameters μ= 0.001; and β=10. When the number of iterations of the algorithm reaches the preset value and the $\nabla E^{\left(k\right)}\left(x\right)$ is less than $g_{min}$ can be used as termination conditions, respectively.

For 30 validation sets, the experimental data show that the average error of the LM-BP neural network is 0.101 °C, the maximum error is 0.488 °C, and the mean square error is 0.003 °C. From the inversion results, it can be seen that the LM algorithm can effectively improve the prediction accuracy of a water temperature inversion BP algorithm.

It can be seen that the LM algorithm needs to control the parameter μ. Switching between the gradient descent method (GDM) and Gauss–Newton method (GNM) also determines that the working mode of the LM algorithm depends on the selection of initial value. When the initial value is not selected properly, the network will easily fall into local minimum after being proficient [33].

#### 2.4.4. PSO-LM-BP Neural Network Algorithm

Through the above experiments, it can be concluded that the advantage of the PSO algorithm is to prevent the BP algorithm from falling into a local minimum, and the advantage of the LM algorithm is to make the BP neural network converge quickly and accurately. The LM algorithm is more dependent on the selection of the initial value, and the PSO algorithm can make up for this shortcoming, so the two algorithms can complement each other. Therefore, the PSO-LM-BP neural network optimization algorithm is given. The algorithm flow chart is as shown in Figure 8 (states 1 and 2 in Figure 7 are described in detail in the block diagram of the LM algorithm).

For the PSO-BP algorithm, the particle population size is set as 20, the update range of velocity is (−5, 5), the update range of position is (−10, 10), and the inertia coefficient decreases linearly from 1 to 0 as the number of iterations increases. For the LM-BP algorithm, the step parameters are μ= 0.001, β=10, the number of iterations is 300, and$\;g_{min}$ is $10^{-10}$. For the PSO-LM-BP algorithm, the PSO algorithm is used to iterate 50 times, and then, the optimal solution of the current population is transferred to the LM algorithm. The optimal solution is obtained when the LM algorithm reaches the ending condition [34,35].

For 30 validation sets, the experimental data show that the average error of the PSO-LM-BP neural network is 0.055 °C, the maximum error is 0.209 °C, and the mean square error is 0.002 °C. From the inversion results, it can be seen that the PSO-LM-BP algorithm can effectively improve the prediction accuracy of the water temperature inversion algorithm.

## 3. Results

In order to evaluate the model, mean square error (MSE) is introduced as evaluation criteria. The calculation formula is as follows:(21)$$MSE\left(\hat{\theta }\right)=\frac{1}{N}\sum_{i=1}^{N}{\left(\hat{\theta }-\theta \right)}^{2}.$$
where $\hat{\theta }$ is the estimated value, and θ is the true value. The results of 30 predictions are shown in Figure 9.

In Figure 9, the loss value is the absolute value of the difference between the actual temperature and the predicted temperature, which shows that the BP neural network algorithm and PSO algorithm have a large prediction error, while the LM algorithm uses second derivative information to make the convergence accuracy higher and the prediction error smaller. However, the LM algorithm needs to adjust the parameter u, which depends on the initial weight matrix. Therefore, in combination with the advantages of the PSO algorithm that can increase the global search capability of the network, this paper uses the PSO algorithm to calculate the initial weight and threshold matrix first, and then, it iterates into the LM algorithm and innovatively proposes the PSO-LM-BP algorithm. Compared with the other three algorithms, the PSO-LM-BP algorithm has higher accuracy and stronger data-fitting ability.

The comparison diagram of the four algorithms is shown in Figure 10a, which shows that the MSE convergence effect of the PSO algorithm is slightly worse, while the convergence process of the other three algorithms is faster and the convergence accuracy is high. Since the details of the other three algorithms cannot be distinguished, the amplification operation is carried out. In Figure 10b, the comparison of MSE values during the convergence of the other three algorithms except for the PSO algorithm shows that by combining the advantages of PSO and LM algorithms, the PSO-LM-BP algorithm improves the iteration efficiency and accuracy, makes it difficult for the network to converge to the local minimum, and reduces the fluctuation of temperature prediction of the broadband correlative microwave radiometer compared with the traditional BP neural network.

The inversion algorithm of the correlative microwave temperature measuring system in this paper is compared with the paper on temperature measurement by a microwave radiometer published by Sun et al. [24]; the results are shown in Table 5.

The PSO algorithm is used to determine the initial network weight parameters due to its global search property, and then, the second derivative information of the LM algorithm is combined to accelerate the convergence speed and improve the iteration efficiency; finally, an innovative PSO-LM-BP high-precision inversion algorithm is proposed in this paper. Compared with other correlative microwave radiometers, in the BP inversion algorithm, the mean square error is reduced by 0.332 °C, the average error is reduced by 0.520 °C, and the maximum error is reduced by 1.149 °C. The high-precision PSO-LM-BP temperature inversion algorithm greatly improves the temperature measurement performance of the correlative microwave radiometer.

## 4. Conclusions

In this paper, a wide-band correlative microwave receiver is designed for non-contact temperature measurement, and then, the correlative microwave receiver is applied to non-contact water temperature measurement. The temperature measurement performance of the receiver is evaluated and analyzed through actual sampling data. The contributions made in this paper and the future discussion of this study are as follows:
A correlative microwave receiver error correction algorithm is designed. A phase correction program is designed to compensate the phase error of the receiver system. The phase error of the signal in each theoretical phase can be minimized. The correction parameters of gain error and misalignment error are obtained by the linear fitting method. After correction, the phase error is reduced from 69.08° to 4.02°.The integral calibration scheme of the correlative microwave receiver’s non-contact temperature measuring system is designed. Using a metal aluminum plate and absorbing material as the calibration source, the calibration scheme of the system is designed according to its brightness temperature and corresponding output voltage. The linear relationship between the output voltage value of the correlative microwave radiometer and the brightness temperature of the object can be obtained accurately.The temperature inversion algorithm of the non-contact temperature measuring system of the correlative microwave radiometer is designed. The influence of the metal aluminum plate, antenna, and transmission line on the output results is considered. The BP neural network algorithm and its improved algorithm PSO-BP, LM-BP, and PSO-LM-BP neural network algorithm are used to obtain the inversion water temperature value. The temperature prediction mean square error of the BP neural network algorithm is 0.629 °C, while the maximum error is 3.351 °C. The PSO algorithm and LM algorithm combined their respective advantages to optimize the back propagation process of the BP neural network algorithm, which improved the inversion accuracy. The mean square error of the PSO-BP algorithm is 0.210 °C, and the mean square error of the LM-BP algorithm is 0.003 °C. The mean square error of the PSO-LM-BP algorithm is 0.002 °C, and the maximum error is 0.209 °C.Laboratory environment, the reflection of the receiver, and the calibration source itself will affect the brightness temperature of the reflection. To improve the precision of temperature measurement, the follow-up works in this paper are as follows: (1) designing a more accurate and stable blackbody calibration source; (2) doing integral calibration operation under a microwave dark room test environment; (3) modifying the integral calibration equation by a correction algorithm.In this paper, the existing data sets of actual measurements are used to design the inversion scheme of the receiver’s non-contact temperature measurement system. However, the maximum error of the PSO-LM-BP inversion algorithm used in this paper is 0.209 K [36,37,38], which is better than that of the existing infrared thermometer 0.90 K. More data sets need to be added in the subsequent work to improve the inversion temperature range and play to the advantages of the neural network algorithm.

## Figures and Tables

**Figure 1 sensors-21-05336-f001:**
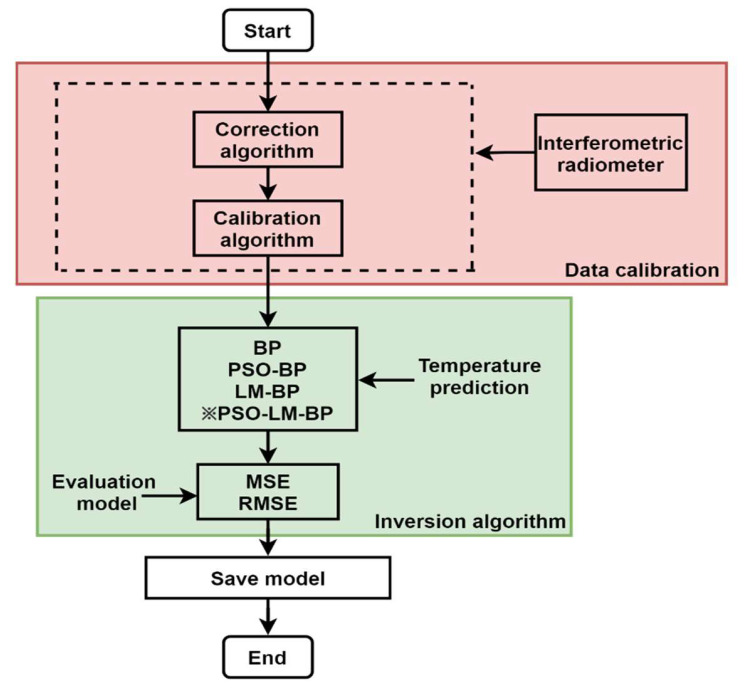
Correlative microwave radiometer calibration and temperature inversion process.

**Figure 2 sensors-21-05336-f002:**
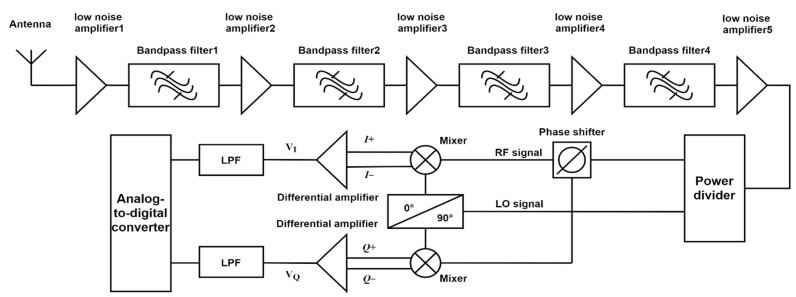
Design block diagram of a broadband microwave receiver.

**Figure 3 sensors-21-05336-f003:**
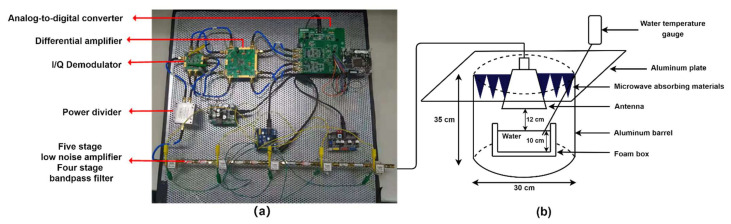
(**a**) Physical picture of broadband correlative microwave receiver; (**b**) Device diagram of non-contact temperature measurement system based on correlative receiver.

**Figure 4 sensors-21-05336-f004:**
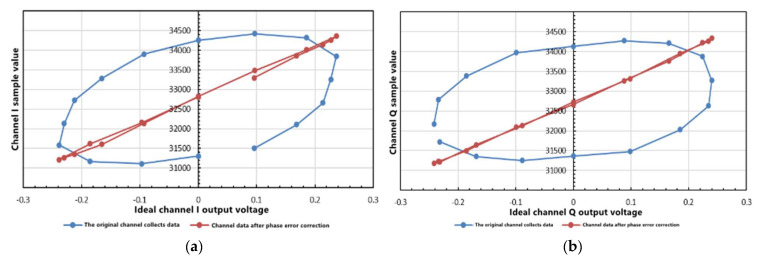
(**a**) The comparison of original sampling data of the I channel before and after correcting the phase error; (**b**) The comparison of the original sampling data of the Q channel before and after correcting the phase error.

**Figure 5 sensors-21-05336-f005:**
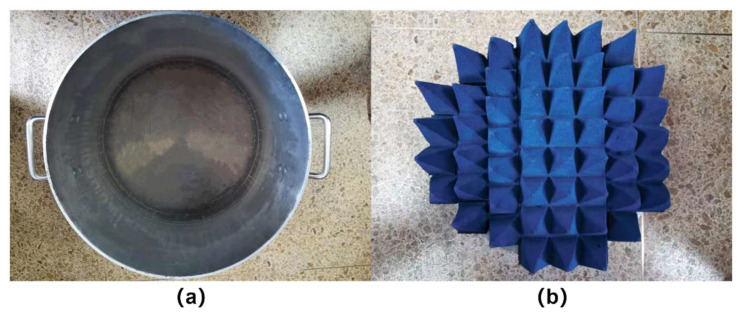
The physical picture of the calibration source. (**a**) Metal aluminum barrel; (**b**) Microwave absorbing material.

**Figure 6 sensors-21-05336-f006:**
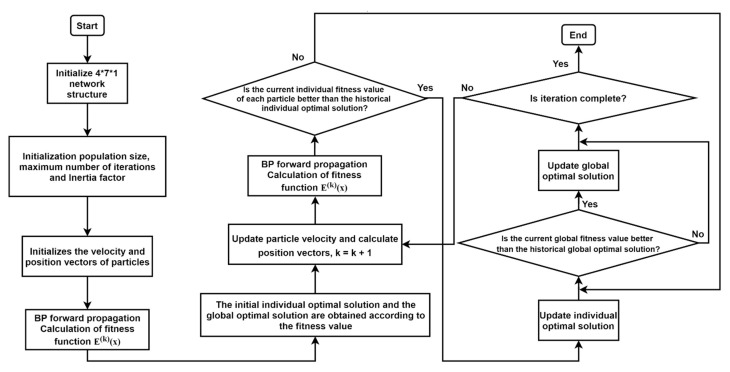
Optimization of BP algorithm flow chart by the particle swarm optimization (PSO) algorithm.

**Figure 7 sensors-21-05336-f007:**
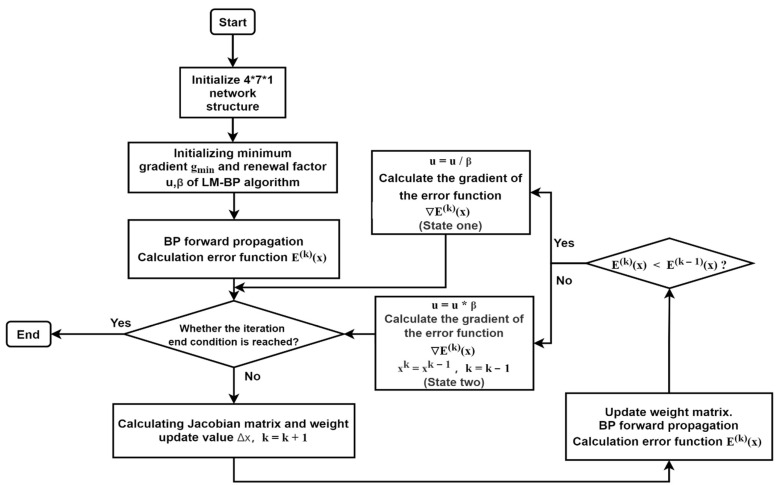
Optimization of the BP algorithm flow chart by the Levenberg–Marquardt (LM) algorithm.

**Figure 8 sensors-21-05336-f008:**
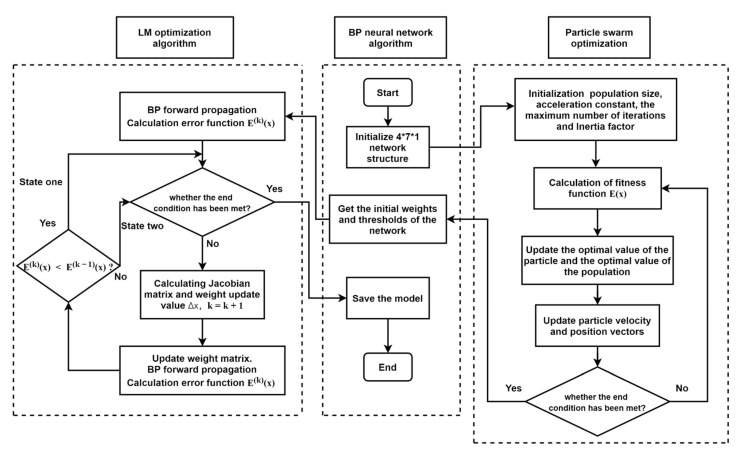
Optimization of the BP algorithm flow chart by particle swarm optimization (PSO) and Levenberg–Marquardt (LM) algorithm.

**Figure 9 sensors-21-05336-f009:**
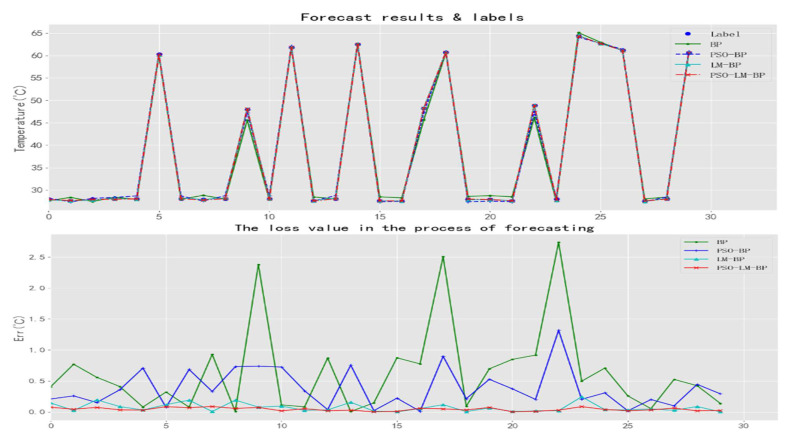
**Up:** Temperature prediction results of four kinds of network prediction processes; **Down:** Loss values of four kinds of network prediction processes.

**Figure 10 sensors-21-05336-f010:**
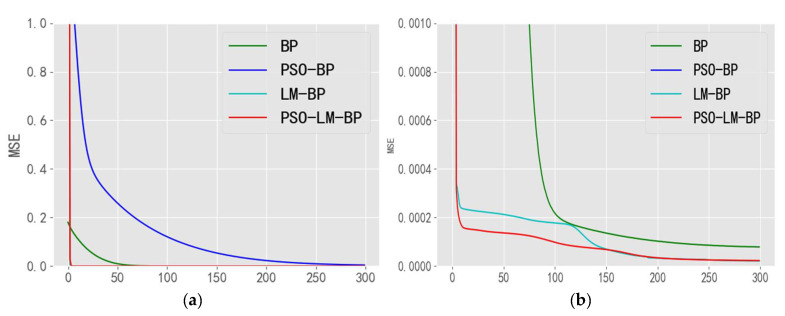
The mean square error values of the training process of the four algorithms. (**a**) The algorithm iterates the overall icon; (**b**) Enlarged local representation.

**Table 1 sensors-21-05336-t001:** Part of the actual experimental measurement results.

Group Number/Serial Number	I Channel Sampling Value	Q Channel Sampling Value	Phase Parameters (°)	Temperature (°C)
1/1	37,595	38,145	−63.558	43.6
1/2	37,443	39,005	−67.606	43.6
1/3	37,629	38,158	−68.407	43.6
1/4	37,869	38,304	−71.950	43.6
1/5	38,533	39,233	−63.617	43.6
1/6	37,987	38,631	−65.363	43.6
2/1	36,892	38,329	−74.692	43.7
2/2	37,257	38,479	−68.716	43.7
2/3	36,955	39,070	−65.056	43.7
2/4	37,461	38,570	−63.076	43.7
2/5	37,874	39,040	−63.823	43.7
2/6	37,842	38,865	−68.656	43.7
3/1	37,036	38,679	−69.342	43.8
3/2	36,679	38,220	−67.701	43.8
3/3	36,977	39,572	−66.907	43.8
3/4	37,919	39,057	−66.771	43.8
3/5	37,110	38,486	−68.692	43.8
3/6	37,293	38,685	−74.338	43.8

**Table 2 sensors-21-05336-t002:** Sampling data during phase scanning.

I Channel Voltage (V)	Q Channel Voltage (V)	Phase Mean ∅ (°)	Theoretical Phase (°)	Phase Error (°)
−0.253	0.003	179.292	247.5	68.208
−0.244	0.095	158.858	225	66.142
−0.180	0.184	134.372	202.5	68.128
−0.097	0.208	114.987	180	65.013
−0.006	0.230	91.590	157.5	65.910
0.079	0.220	70.362	135	64.638
0.173	0.169	44.334	112.5	68.166
0.227	0.077	18.702	90	71.298
0.253	−0.021	−4.743	67.5	72.2431
0.237	−0.112	−25.407	45	70.407
0.164	−0.197	−50.120	22.5	72.620
0.075	−0.214	−70.638	0	70.638
−0.016	−0.231	−94.000	−22.5	71.500
−0.100	−0.216	−114.869	−45	69.869
−0.193	−0.160	−140.313	−67.5	72.813
−0.224	−0.092	−157.698	−90	67.698

**Table 3 sensors-21-05336-t003:** Data after error correction.

I Channel Voltage (V)	Q Channel Voltage (V)	Phase Mean ∅ (°)	Theoretical Phase (°)	Phase Error (°)
−0.114	−0.240	−115.401	247.5	2.901
−0.197	−0.200	−134.614	225	−0.386
−0.258	−0.108	−157.386	202.5	−0.114
−0.251	−0.021	−175.12	180	−4.880
−0.239	0.071	163.513	157.5	−6.013
−0.200	0.146	143.764	135	−8.764
−0.118	0.216	118.667	112.5	−6.167
−0.012	0.234	93.033	90	−3.033
0.089	0.222	68.264	67.5	−0.764
0.169	0.174	45.953	45	−0.953
0.222	0.077	19.072	22.5	3.428
0.207	−0.012	−3.454	0	3.454
0.190	−0.104	−28.625	−22.5	6.125
0.146	−0.177	−50.403	−45	5.403
0.060	−0.243	−76.023	−67.5	8.523
−0.015	−0.247	−93.389	−90	3.389

**Table 4 sensors-21-05336-t004:** Calibration data.

Measured Object	Temperature T (°C)	Emissivity ε	$\mathrm{Brightness}\;\mathrm{Temperature}\;\;T_{B\;}(\text{°}C)$	Correction Voltage V (V)
Metal aluminum barrel	27	0.26	7.020	1.213
Microwave absorbing materials	28.4	0.995	28.258	1.255

**Table 5 sensors-21-05336-t005:** Comparison of the temperature inversion algorithm with another correlative microwave radiometer.

Temperature Inversion Algorithm	Mean Square Error (°C)	Average Error (°C)	Maximum Error (°C)
Multiple linear regression algorithm [24]	0.607	0.759	1.777
BP neural network algorithm [24]	0.334	0.575	1.358
BP neural network algorithm	0.629	0.824	3.351
PSO-BP algorithm	0.210	0.521	2.920
LM-BP algorithm	0.003	0.101	0.488
PSO-LM-BP algorithm	0.002	0.055	0.209

## Data Availability

Not applicable.

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
