# Peer review of "High-Precision Temperature Inversion Algorithm for Correlative Microwave Radiometer"

_sensors, 2021, doi:10.3390/s21165336_

Round 1

Reviewer 1 Report

I have added comments in the pdf

Author Response

Dear Reviewer 1,‎

Thank you for your report. ‎

We would also like to thank you, for taking the time to read our paper and for your important comments and remarks.

Yours sincerely

‎Kai Zhang

Reviewer 2 Report

In general, there are two questions to be checked or verified  in principle for further justifying the results of the manuscript:

1.The calibration of the microwave radiometer  using metal and absorption load. The brightness temperature from the metal is not only from the emisson from itself, but also from the reflection from the surrounding, esp. including the refrected radiation from the radiometer itself. In the calibration, only the emission of the metal is used for calibrating the receiver, and the reflection of the metal is not included, which is more important in determining the brightness temperature from metal and the real brightness temperature from metal is far more than a few Kelvin  at the configuration of the test, so the calibration method is wrong and the results is not convincible.

2. Microwave radiometer can detect the physical temperature through retrieving methods from measured bightness temperature, but not measure directly the physical temperatue! One of other  important factors is dielectric constant of the target, esp. for targets contenting water. There are some mistakens  on the concepts of  microwave radiometer applications.

Author Response

Dear Reviewer 2,‎

Thank you for your report. ‎

We would also like to thank you, for taking the time to read our paper and for your important comments and remarks.

Yours sincerely

‎Kai Zhang

Round 2

Reviewer 2 Report

Comments as follows:

Majors:

1.A more detailed introduction on the receiver system should be given following Figure 2 (line 92), such as on receiver, antenna, or some necessary specifications of system, etc. which is necessary for learning about the test and verifying the results.

2.The contents from line 101 to 115 should be moved to the “1. introduction ”. The concept of calibration in this paragraph is not properly defined.

3.Line 120, It seems that ' j 'in formula (2) is not be kept here since VQ is already defined as voltage.

4.The description on ‘Calibration based on the temperature measuring system’ is not clear, and some concept is not defined properly. In fact the content from line 177 to 230 are not necessary listed here since there is no special information on calibration used for the following test but a two-point calibration method widely applied in microwave radiometer calibration.

5.What is one of the calibration source “Metal aluminum barrel”, there is no introduction at all at its first shown at line 204 to 206. And how do you give an emissivity of 0.26 and brightness temperature of 7.020. Moreover, The unit of brightness temperature should be in Kelvin, but not in degree C!

  1. Section 2.4 is poorly organized and the contents in this section are too tediously long, which should be shortened and refined clearly and logically.
  2. The sentence at line 250 and Figure 6 are not relevant to "inversion", which should be moved to introduction section if necessary.

Minors

See  notes on the PDF version in the attached file.

Author Response

(The authors gave the same response as above.)
